# Safety assessment of sodium zirconium cyclosilicate: A FAERS-based disproportionality analysis

Yongfei Yu[1,2,3☯], Kaiyu Zhang[1,2,3☯], Jinglin Gao[1,2,3], Guoshun Huang[1,2,3], Chen Yong[1,2,3], Yuan Wei[1,2,3], Enchao Zhou[1,2,3]*

1 Jiangsu University Key Laboratory of TCM Nourishing Kidney Essence and Anti-Aging Research, The First Clinical Medical College, Nanjing University of Chinese Medicine, Nanjing, China, 2 Division of Nephrology, Affiliated Hospital of Nanjing University of Chinese Medicine, Jiangsu Province Hospital of Chinese Medicine, Nanjing, China, 3 Zou's Nephrology Medicine Intangible Cultural Heritage Inheritance Studio, Nanjing, China

☯ These authors contributed equally to this work.
* zhouenchao@njucm.edu.cn

## Abstract

Sodium zirconium cyclosilicate (SZC) is a novel therapeutic agent for hyperkalemia. However, limited reports have described its real-world safety. In this study, we used the Food and Drug Administration Adverse Event Reporting System (FAERS) database to evaluate the safety profilet of SZC. This retrospective analysis involved FAERS data from the third quarter of 2018 to the fourth quarter of 2023. A disproportionality analysis, including reporting odds ratio (ROR), proportional reporting ratio (PRR), empirical bayesian geometric mean(EBGM), and bayesian confidence propagation neural network (BCPNN), was conducted to quantify the signals of SZC-related adverse events (AEs). Sex subgroup analyses were also conducted. A total of 1,154 SZC-related AE reports were obtained from the FAERS database. SZC-related AEs primarily targeted 21 system organ classes after simultaneously conforming to the four algorithms. The most common AEs included X-ray gastrointestinal tract abnormalities(ROR = 3888.02[1051.8, 14372.15]) and blood potassium abnormal(ROR = 212.53[134.73, 335.26]), consistent with prior findings and clinical trials. Unexpected significant AEs included ileus (ROR = 27.82[14.44, 53.59]), death (ROR = 17.49[15.7, 19.49]) and congestive cardiac failure (ROR = 16.66[10.94, 25.37]). Sex-based differences were observed for AEs. The median onset time of SZC-related AEs was 33 days. This study confirms common AEs of SZC and identifies new ones, highlighting the need for careful monitoring and facilitating its safe use in clinical settings. However, disproportionality analysis cannot formally prove causation and further studies are needed to confirm these associations.

## Introduction

Hyperkalemia is defined as a potassium concentration $\geq 5.5$ mmol/L; it is a common electrolyte disorder [1]. It often arises during medical conditions, including chronic kidney disease (CKD), heart failure, diabetes, and severe tissue trauma. Additionally, the use of medications,

**Data availability statement:** All relevant data are within the paper and FAERS database. The data have been made fully public for anyone to view. https://fis.fda.gov/extensions/FPD-QDE-FAERS/FPD-QDE-FAERS.html.

**Funding:** Our research was supported by the National Natural Science Foundation of China under grant 82474427 to EZ, Jiangsu Natural Science Foundation of China under grant BE2023791 to EZ, Jiangsu Province Leading Talents Cultivation Project for Traditional Chinese Medicine under grant SLJ0319 to EZ, Jiangsu Province Traditional Chinese Medicine Science and Technology Project under grant ZD202007 to EZ and Postgraduate Research & Practice Innovation Program of Jiangsu Province under grant SJCX22_0724 to YY. The funders had no role in study design, data collection and analysis, decision to publish, or preparation of the manuscript.

**Competing interests:** The authors have declared that no competing interests exist.

such as renin-angiotensin-aldosterone system (RAAS) inhibitors, mineralocorticoid receptor antagonists, and nonsteroidal anti-inflammatory drugs, has been strongly associated with hyperkalemia onset [2]. Hyperkalemia can lead to both cardiac hyperexcitability (ventricular tachycardia and ventricular fibrillation) and suppression (bradycardia, atrioventricular block, intraventricular conduction delay, and cardiac arrest), all of which can be life-threatening [3]. The severity and duration of hyperkalemia have been strongly correlated with adverse events (AEs) and increased mortality among critically ill patients [4].

Sodium zirconium cyclosilicate (SZC) is a novel therapeutic agent for hyperkalemia. It is a non-absorbable, non-polymer zirconium silicate compound that selectively exchanges hydrogen and sodium ions for potassium and ammonium ions in the gastrointestinal tract. Thus, SZC enhances fecal potassium excretion and effectively lowers serum potassium levels [5]. SZC effectively manages hyperkalemia; nonetheless, some AEs have been associated with its clinical use [6]. Insufficient systematic and comprehensive research, particularly those leveraging real-world data and large datasets, has described SZC-related AEs.

In the US, the Food and Drug Administration Adverse Event Reporting System (FAERS) is a publicly accessible, spontaneous reporting database. It compiles post-marketing safety reports from healthcare professionals, consumers, and manufacturers. The FAERS is key to detecting AEs associated with pharmaceutical interventions [7,8], facilitating the early identification of drug safety concerns. Disproportionality analysis of AE signals in the FAERS can help identify and manage unrecognized safety signals during clinical development to enhance risk management strategies [9].

In this study, we retrospectively analyzed SZC-related AEs through data mining of the FAERS from the third quarter of 2018 to the fourth quarter of 2023. We aimed to identify new AE signals and clarify the safety profile of SZC. Our findings will optimize clinical practice, promote the safe therapeutic use of SZC, and safeguard patient health.

## Materials and methods

### Data source and data processing

The FAERS database aggregates AEs, allowing researchers to identify safety signals and assess the link between drug dosage and AEs [10]. The system is open to healthcare providers, consumers, and manufacturers globally. The FAERS dataset consists of seven components: DEMO (patient demographics and administrative information), DRUG (details of the drugs involved), REAC (coded adverse events), OUTC (patient outcomes), RPSR (report sources), THER (therapy start and stop dates), and INDI (indications for drug administration).

Individual case safety reports from the FAERS between the third quarter of 2018 and the fourth quarter of 2023 were utilized. All data used in this retrospective study were fully anonymized and were obtained from publicly available databases. All authors had no access to information that could identify individual participants during or after data collection. Because of duplicate or non-standardized reports, raw data were cleaned to ensure accuracy. The drug name was standardized through the Medex_UIMA_1.8.3 system. Reports suspecting Sodium zirconium cyclosilicate as the primary drug associated with AEs were extracted. The FDA-recommended criteria were followed to remove duplicate entries [11,12]. For identical CASEIDs, the latest FDA_DT was selected; for similar CASEID and FDA_DT, the record with the higher PRIMARYID was selected [13].

### Study design and statistical analysis

The Medical Dictionary for Regulatory Activities (MedDRA) is a global standardized terminology used to streamline AEs [14]. Its hierarchical structure includes multiple levels, with the

system organ class (SOC) being the highest, used for classifying AEs in drug safety systems [15]. To systematically analyze AEs, we coded them using preferred terms (PTs) and mapped them to their respective SOC categories in MedDRA (version 26.0) [16]. Key outcomes included death (DE), life-threatening events (LT), hospitalization (HO), disability (DS), congenital anomalies (CA), or other significant conditions (OT) [17]. Clinical characteristics such as sex, age, and reporting countries were also compiled.

In pharmacovigilance, disproportionality analysis is essential for identifying drug-related AE signals [18]. To analyze AEs linked to SZC, we used four methods: Reporting Odds Ratio (ROR), Proportional Reporting Ratio (PRR), Bayesian Confidence Propagation Neural Network (BCPNN), and Empirical Bayesian Geometric Mean (EBGM) [19]. This multi-method approach addresses the limitations of individual algorithms, improving data mining precision and reliability. Detailed formulas and thresholds are provided in S1 and S2 Tables. To ensure the robustness of our findings, separate sex-based disproportionality analyses were conducted. The data predominantly focused on patients aged ≥ 65 years; thus, age was not analyzed as a separate subgroup.

Additionally, the time to onset (TTO) and the likelihood of severe AE outcomes were calculated. TTO was defined as the time interval between the date of initiating SZC therapy (START_DT) and that of AE onset (EVENT_DT). After initiating SZC therapy, the frequency of AEs depends on the mechanism of action of SZC and may change gradually. Contrarily, AEs unrelated to drug therapy tend to occur consistently [20]. A comprehensive TTO analysis using medians, quartiles, extremes, and the Weibull distribution test was conducted [21]. The Weibull Proportionality Test is widely applied to model time-to-event data. It allows assessing and predicting gradual shifts in the risk incidence, with the scale ($\alpha$) and shape ($\beta$) parameters characterizing the Weibull distribution form [13,22]. This analysis facilitated an understanding of the variations in AE incidence after administering SZC.

All data processing and statistical analyses were performed using R software version 4.1.3 (https://www.r-project.org/) and and the data were organized using Microsoft Excel 2021. The analysis results of plots were generated using the CNSknowall platform (https://cnsk-nowall.com), a comprehensive web service for data analysis and visualization. Fig 1 presents a detailed flowchart outlining the step-by-step process of data extraction, cleaning, and analysis.

## Results

### Population characteristics

From the third quarter of 2018 to the fourth quarter of 2023, 8,067,923 AE reports were collected from the FAERS database. After rigorous data cleaning, 1,154 SZC-related AEs were identified for an in-depth analysis. Table 1 summarizes the demographic characteristics of the patients who experienced SZC-related AEs. AEs were more commonly reported in men (48.44%) than in women (29.03%). Older adults (age ≥ 65 years) accounted for 39.26% of the cases. Most AE reports originated from the United States (76.60%), followed by Japan (17.07%), China (1.39%), the United Kingdom (1.39%), and Colombia (0.87%). AEs were predominantly reported by physicians (33.97%), consumers (30.59%), and pharmacists (29.29%). A total of 481 cases reported death as the most frequent AE (51.61%), followed by 224 cases of other serious events (24.03%), 184 cases of hospitalization (19.74%), and 34 cases of life-threatening events (3.65%). Following the introduction of SZC in May 2018, the trend indicates a yearly increase in the number of SZC-related AEs, with most AEs recorded in 2023 (n = 419).

S3 Table summarizes the SZC signal strengths and reporting patterns at the System Organ Class (SOC) level. Notably, SZC-related AEs encompassed 21 organ systems. Fig 2A illustrates

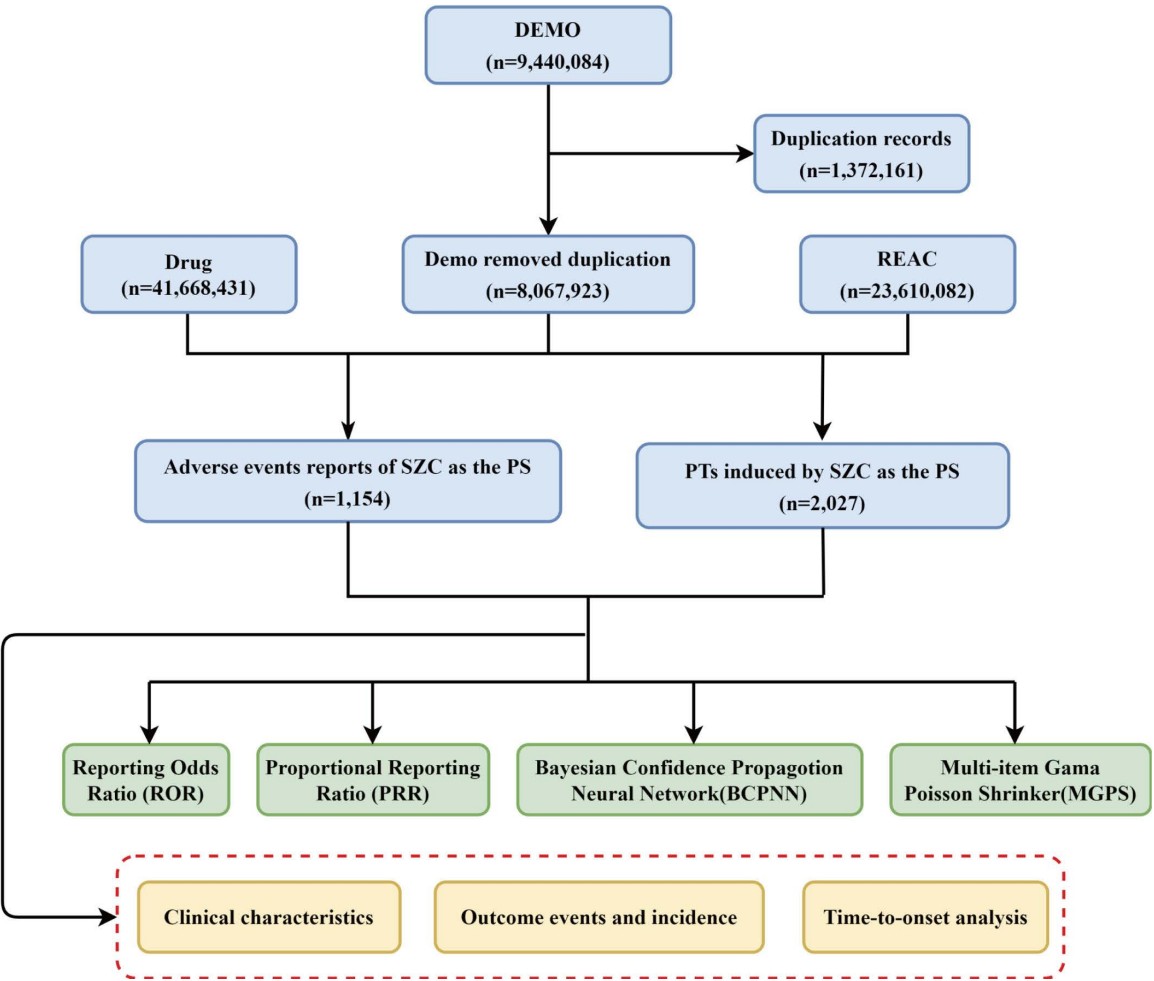

**Fig 1. The flowchat of extracting and analyzing SZC-related AEs from FAERS database.** SZC, sodium zirconium cyclosilicate; AEs, adverse events; FAERS, Food and Drug Administration Adverse Event Reporting System.

AE distribution across SZC-related SOCs. Fig 2B illustrates the reporting odds ratio (ROR) and 95% confidence intervals (CI), reflecting the signal strength for SZC-related SOCs.

The following SOC categories were most strongly associated with SZC therapy: general disorders and administration site conditions (n = 620, ROR 1.97 [1.79, 2.16], PRR 1.67 [1.57, 1.77], IC025 0.61), gastrointestinal disorders (n = 324, ROR 2.11 [1.87, 2.38], PRR 1.93 [1.75, 2.13], IC025 0.78), investigations (n = 250, ROR 2.19 [1.92, 2.50], PRR 2.04 [1.81, 2.29], IC025 0.84), metabolism and nutrition disorders (n = 115, ROR 2.91 [2.41, 3.51], PRR 2.8 [2.35, 3.34], IC025 1.22, EBGM05 2.39), cardiac disorders (n = 107, ROR 2.61 [2.15, 3.17], PRR 2.53 [2.12, 3.02], IC025 1.06, EBGM05 2.15) and renal and urinary disorders (n = 250, ROR 1.70 [1.34, 2.15], PRR 1.67 [1.35, 2.07], IC025 0.41).

## Preferred term signals and subgroup analyses

A signal detection analysis identified 39 significantly disproportionate preferred terms (PTs), which simultaneously met the criteria of all four signal detection methods. Fig 3 describe the leading 30 PTs, ranked by the number of AE reports and the strength of AE signals. The details were shown in Table 2. Commonly reported AEs in patients who underwent SZC therapy

**Table 1.** Clinical characteristics of reports of SZC from the FAERS database.

| Characteristics | Case Number, n | Case proportion(%) |
|---|---|---|
| Number of events | 1,154 | |
| **Sex** | | |
| Female | 335 | 29.03 |
| Male | 559 | 48.44 |
| Unknown | 260 | 22.53 |
| **Age (year)** | | |
| <45 | 28 | 2.43 |
| 45-64 | 106 | 9.19 |
| 65-74 | 120 | 10.40 |
| >74 | 333 | 28.86 |
| Unknown | 567 | 49.13 |
| **Serious outcome** | | |
| Death | 481 | 51.61 |
| Hospitalization | 184 | 19.74 |
| Life threatening | 34 | 3.65 |
| Disability | 8 | 0.86 |
| Required intervention | 1 | 0.11 |
| Other | 224 | 24.03 |
| **Reporter** | | |
| Physician | 392 | 33.97 |
| Consumer | 353 | 30.59 |
| Pharmacist | 338 | 29.29 |
| Other health-professional | 5 | 0.43 |
| Unknown | 66 | 5.72 |
| **Reported region** | | |
| United States | 884 | 76.6 |
| Japan | 197 | 17.07 |
| China | 16 | 1.39 |
| United Kingdom | 16 | 1.39 |
| Colombia | 10 | 0.87 |
| Other | 31 | 2.69 |
| **Reporting Year** | | |
| 2018 | 7 | 0.61 |
| 2019 | 100 | 8.67 |
| 2020 | 157 | 13.60 |
| 2021 | 202 | 17.50 |
| 2022 | 269 | 23.31 |
| 2023 | 419 | 36.31 |

SZC, sodium zirconium cyclosilicate; FAERS, Food and Drug Administration Adverse Event Reporting System.

Signal detection at the System Organ Class level.

included those already listed on the drug label, such as abnormal gastrointestinal X-rays or computed tomography (CT) scans, blood potassium abnormalities, edema, constipation, and intestinal perforation. Moreover, unexpected AEs not mentioned in the SZC drug label, including ileus (n = 9, ROR 27.82), death (n = 415, ROR 17.49), congestive cardiac failure (n = 22, ROR 16.66), and metabolic acidosis (n = 10, ROR 9.18), were identified.

**A**

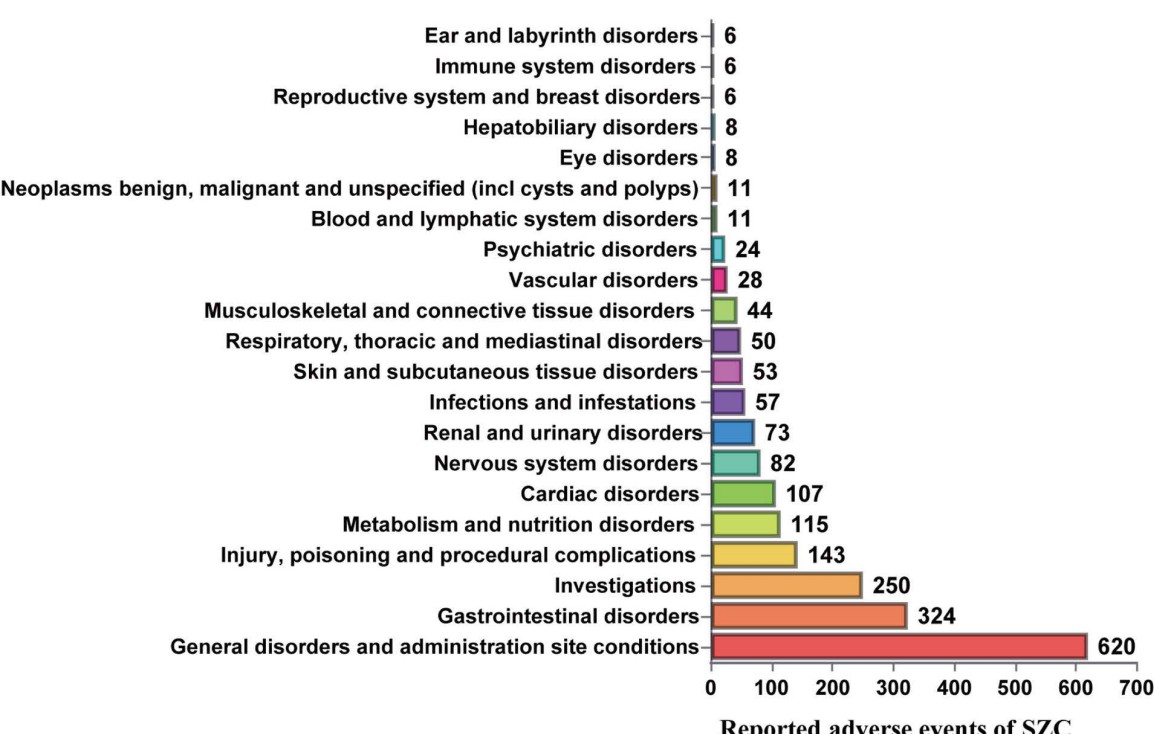

**B**

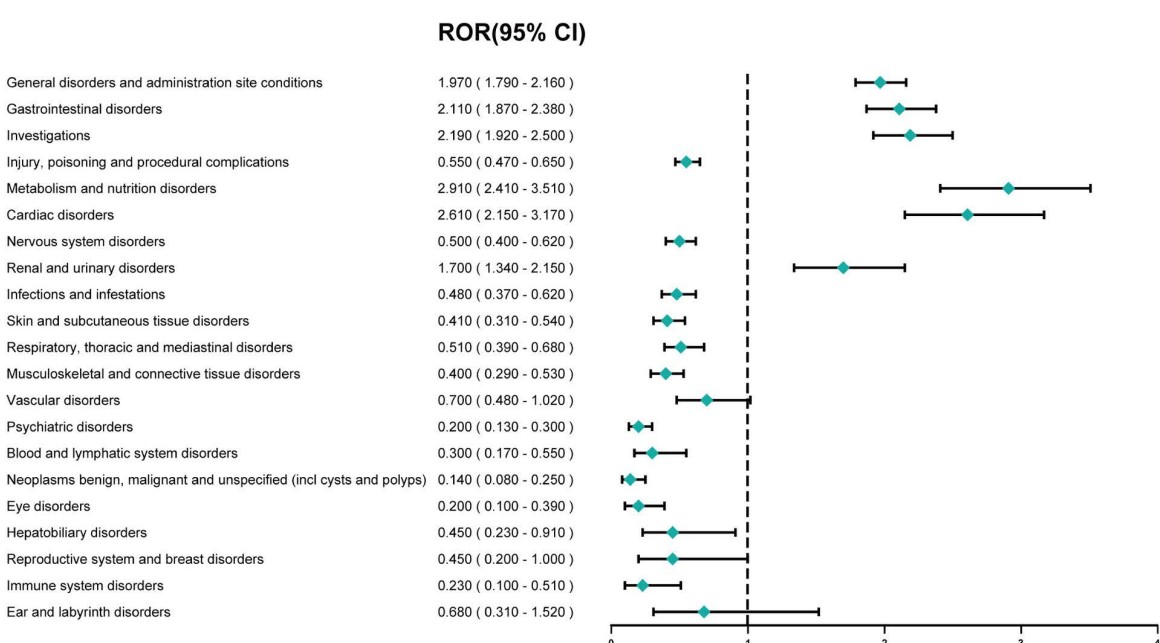

**Fig 2. Signals detection at the SOC level.** (A) The bar chart displays the reported cases of AEs at each SOC level. (B) Signals detection at the SOC level. The ROR values and their 95% confidence intervals (95% CI) are visualized. SOC, system organ class; AEs, adverse events; ROR, reporting odds ratio; SZC, sodium zirconium cyclosilicate.

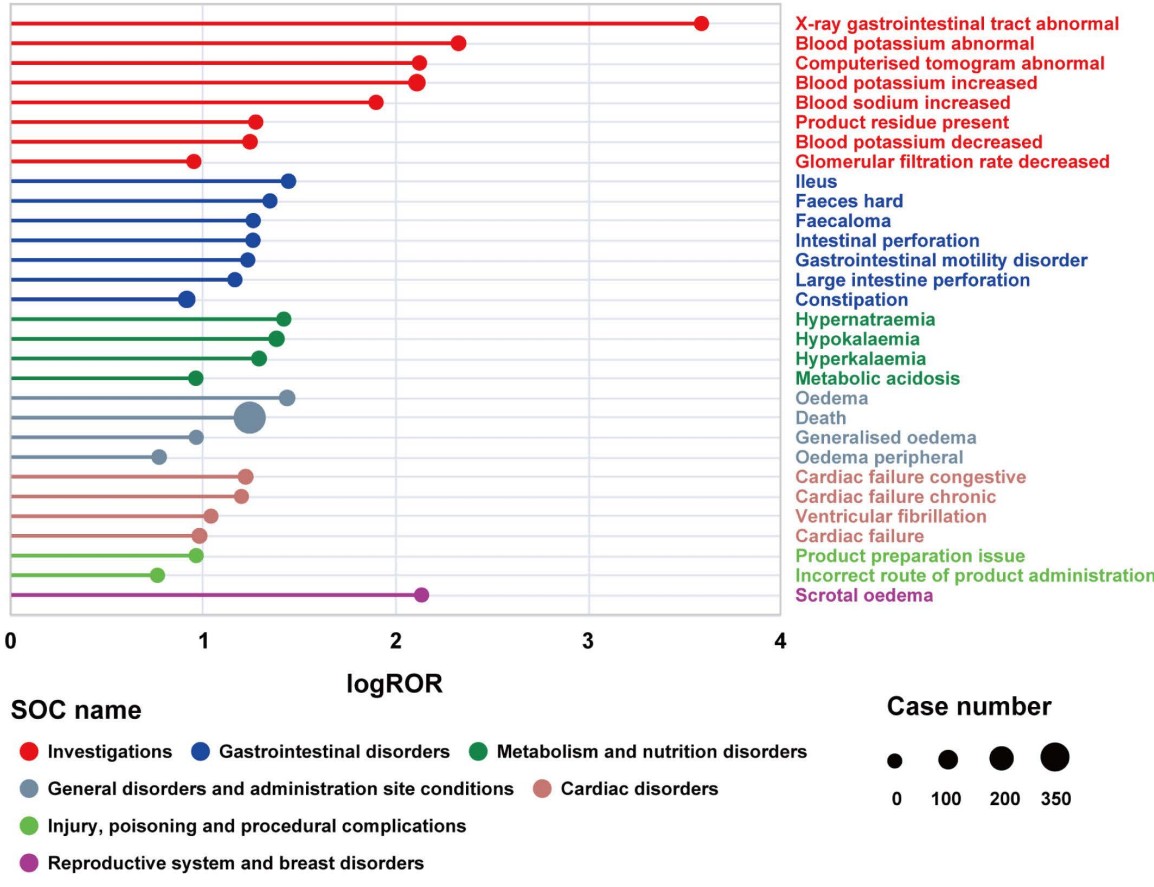

**Fig 3. Signals detection at the PT level.** PT, preferred term.

To explore the influence of sex on SZC-related AEs, subgroup analyses were conducted. PTs were compared between men and women(Fig 4). Some AEs, such as scrotal edema, edema, hypokalemia, and ileus, were more prevalent in men (S4 Table). In women, high-risk AEs included elevated blood sodium levels, fecaloma, congestive heart failure, and decreased glomerular filtration rate (GFR)(S5 Table).

## Time-to-onset analysis

After excluding inaccurate, missing, or unknown records, time-to-onset data for SZC-related AEs were available for 240 reports. The median onset time was 33 days (interquartile range [IQR] 5–167 days). Most AEs (n = 117, 48.75%) occurred within the first month of SZC therapy (Fig 5). Evaluation of the Weibull Shape Parameter analysis (Table 3) suggested a calculated shape parameter (β) of 0.62 (95% CI 0.54–0.70). The result suggested a tendency towards early failure. However, in 12.08% of the cases, the AEs were reported even after 1 year of SZC therapy, highlighting the potential for long-term risks.

## Discussion

To the best of our knowledge, this is the first pharmacovigilance study to analyze the real-world safety profile of SZC using data from the FAERS. Our findings characterize SZC-related AEs and highlight major AEs that are not listed on the SZC drug label.

**Table 2. Signal strength of top 30 AEs of SZC at the PT level in FAERS database.**

| SOC | PT | N | ROR (95% CI) | PRR (95% CI) | $\chi^2$ | IC (IC025) | EBGM (EBGM05) |
|------|-----|---|---------------|---------------|----------|-------------|----------------|
| **Investigations** | X-ray gastrointestinal tract abnormal | 3 | 3888.02 (1051.8, 14372.15) | 3882.27 (1044.17, 14434.48) | 8730.6 | 11.51 (9.89) | 2911.95 (975.19) |
| | Blood potassium abnormal | 19 | 212.53 (134.73, 335.26) | 210.55 (134.15, 330.47) | 3892.37 | 7.69 (7.05) | 206.83 (141.25) |
| | Computerised tomogram abnormal | 7 | 133.24 (63.17, 281.02) | 132.78 (63.05, 279.64) | 905.22 | 7.04 (6.03) | 131.3 (70.32) |
| | Blood potassium increased | 59 | 129.24 (99.62, 167.67) | 125.51 (97.28, 161.93) | 7211.49 | 6.96 (6.58) | 124.18 (99.88) |
| | Blood sodium increased | 6 | 79.28 (35.48, 177.16) | 79.05 (35.39, 176.56) | 459.28 | 6.3 (5.22) | 78.52 (40.07) |
| | Product residue present | 6 | 18.8 (8.43, 41.92) | 18.74 (8.39, 41.86) | 100.64 | 4.23 (3.15) | 18.72 (9.57) |
| | Blood potassium decreased | 16 | 17.55 (10.72, 28.71) | 17.42 (10.67, 28.43) | 247.31 | 4.12 (3.43) | 17.39 (11.52) |
| | Glomerular filtration rate decreased | 4 | 8.97 (3.36, 23.94) | 8.96 (3.36, 23.87) | 28.26 | 3.16 (1.9) | 8.95 (3.94) |
| **Gastrointestinal disorders**79.5 pt | Ileus | 9 | 27.82 (14.44, 53.59) | 27.7 (14.51, 52.89) | 231.12 | 4.79 (3.89) | 27.64 (15.97) |
| | Faeces hard | 3 | 22.29 (7.17, 69.24) | 22.26 (7.14, 69.38) | 60.79 | 4.47 (3.06) | 22.21 (8.6) |
| | Faecaloma | 3 | 18.24 (5.87, 56.66) | 18.22 (5.85, 56.79) | 48.74 | 4.19 (2.77) | 18.19 (7.05) |
| | Intestinal perforation | 6 | 18.19 (8.16, 40.56) | 18.14 (8.12, 40.52) | 97.02 | 4.18 (3.11) | 18.11 (9.26) |
| | Gastrointestinal motility disorder | 3 | 17.08 (5.5, 53.04) | 17.05 (5.47, 53.14) | 45.27 | 4.09 (2.67) | 17.03 (6.6) |
| | Large intestine perforation | 3 | 14.65 (4.72, 45.48) | 14.63 (4.69, 45.6) | 38.04 | 3.87 (2.45) | 14.61 (5.66) |
| | Constipation | 58 | 8.24 (6.35, 10.71) | 8.04 (6.23, 10.37) | 358.4 | 3.01 (2.63) | 8.03 (6.46) |
| **Metabolism and nutrition disorders** | Hypernatraemia | 4 | 26.3 (9.85, 70.21) | 26.25 (9.85, 69.94) | 96.93 | 4.71 (3.44) | 26.19 (11.51) |
| | Hypokalaemia | 35 | 24.11 (17.25, 33.69) | 23.71 (16.99, 33.09) | 760.32 | 4.56 (4.09) | 23.66 (17.89) |
| | Hyperkalaemia | 21 | 19.55 (12.71, 30.05) | 19.35 (12.57, 29.78) | 365.09 | 4.27 (3.67) | 19.32 (13.48) |
| | Metabolic acidosis | 10 | 9.18 (4.93, 17.1) | 9.14 (4.88, 17.11) | 72.51 | 3.19 (2.34) | 9.14 (5.43) |
| **General disorders and administration site conditions** | Oedema | 40 | 27.36 (20, 37.43) | 26.84 (19.61, 36.73) | 993.5 | 4.74 (4.3) | 26.78 (20.6) |
| | Death | 415 | 17.49 (15.7, 19.49) | 14.12 (13.06, 15.27) | 5126.5 | 3.82 (3.67) | 14.1 (12.88) |
| | Generalised oedema | 3 | 9.23 (2.97, 28.66) | 9.22 (2.96, 28.74) | 21.97 | 3.2 (1.79) | 9.21 (3.57) |
| | Oedema peripheral | 16 | 5.92 (3.62, 9.68) | 5.88 (3.6, 9.6) | 64.84 | 2.55 (1.87) | 5.88 (3.89) |
| **Cardiac disorders** | Cardiac failure congestive | 22 | 16.66 (10.94, 25.37) | 16.49 (10.93, 24.89) | 319.91 | 4.04 (3.45) | 16.47 (11.59) |
| | Cardiac failure chronic | 3 | 15.84 (5.1, 49.19) | 15.82 (5.08, 49.31) | 41.59 | 3.98 (2.57) | 15.8 (6.12) |
| | Ventricular fibrillation | 3 | 11 (3.54, 34.16) | 10.99 (3.53, 34.25) | 27.21 | 3.46 (2.04) | 10.98 (4.25) |
| | Cardiac failure | 25 | 9.59 (6.46, 14.22) | 9.48 (6.41, 14.03) | 189.72 | 3.24 (2.69) | 9.47 (6.81) |
| **Injury, poisoning and procedural complications** | Product preparation issue | 3 | 9.22 (2.97, 28.64) | 9.21 (2.95, 28.71) | 21.95 | 3.2 (1.79) | 9.21 (3.57) |
| | Incorrect route of product administration | 6 | 5.8 (2.6, 12.92) | 5.78 (2.59, 12.91) | 23.73 | 2.53 (1.46) | 5.78 (2.96) |
| **Reproductive system and breast disorders** | Scrotal oedema | 3 | 136.69 (43.76, 426.98) | 136.49 (43.79, 425.41) | 398.81 | 7.08 (5.65) | 134.92 (52.02) |

AEs, adverse events; SZC, sodium zirconium cyclosilicate; SOC, system organ class; PT, preferred term; ROR, reporting odds ratio; PRR, proportional reporting ratio; BCPNN, bayesian confidence propagation neural network; CI, confidence interval; χ2, chi-squared; IC, information component; IC025, the lower bound of 95% CI; EBGM, empirical Bayesian geometric mean; EBGM05, the lower bound of 95% CI; N, the number of reports.

These findings facilitate an understanding of the SZC risk profile, which can serve as a reference for rationally administering this therapy. The annual trends in SZC-related reports increased from 2018 to 2023, suggesting a continued increase in SZC reports in the future.

Disproportionality analysis indicated significant signal strengths at the SOC level for categories, such as metabolism and nutrition disorders, cardiac disorders, investigations, gastrointestinal disorders, and general disorders and administration site conditions. The signals

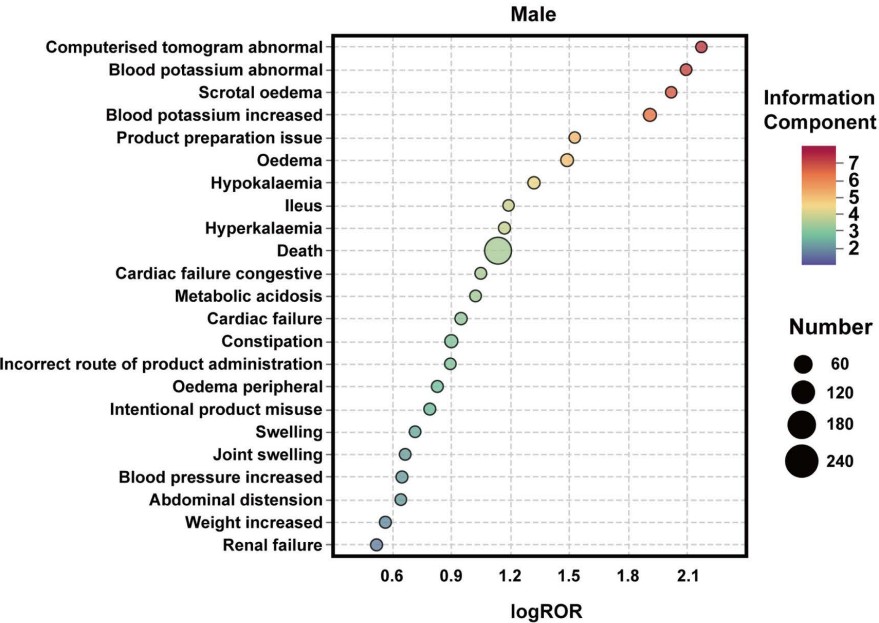

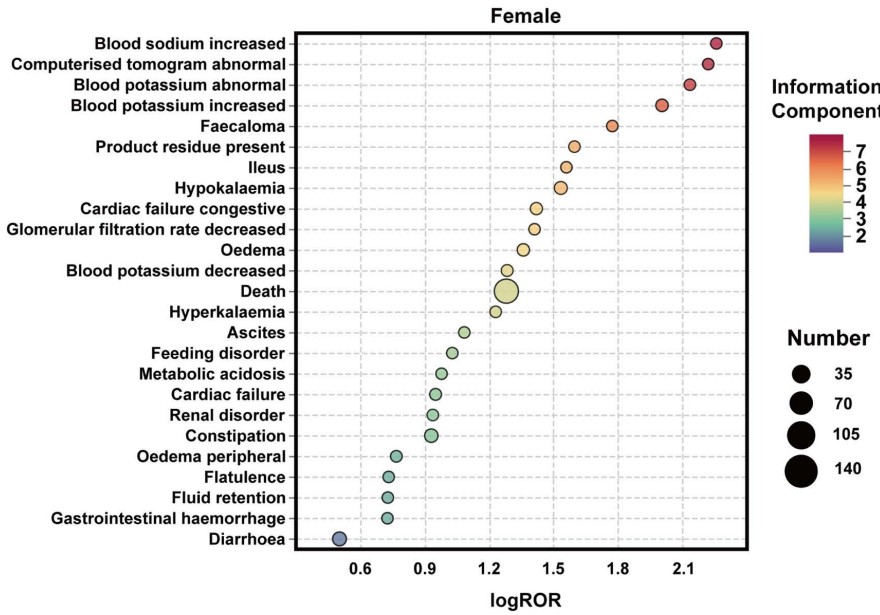

**Fig 4. Subgroup analysis based on sex for SZC related AEs.** (A) Male group; (B) Female group. SZC, sodium zirconium cyclosilicate; AEs, adverse events; ROR, reporting odds ratio; log ROR, logarithm of the reporting ROR; AEs, adverse events.

associated with metabolism and nutrition disorders, general disorders, and administration site conditions were consistent with safety data from the drug label and clinical trials. However, significant signals were obtained for certain SOCs with fewer reports, such as cardiac disorders (n = 107), warranting further attention.

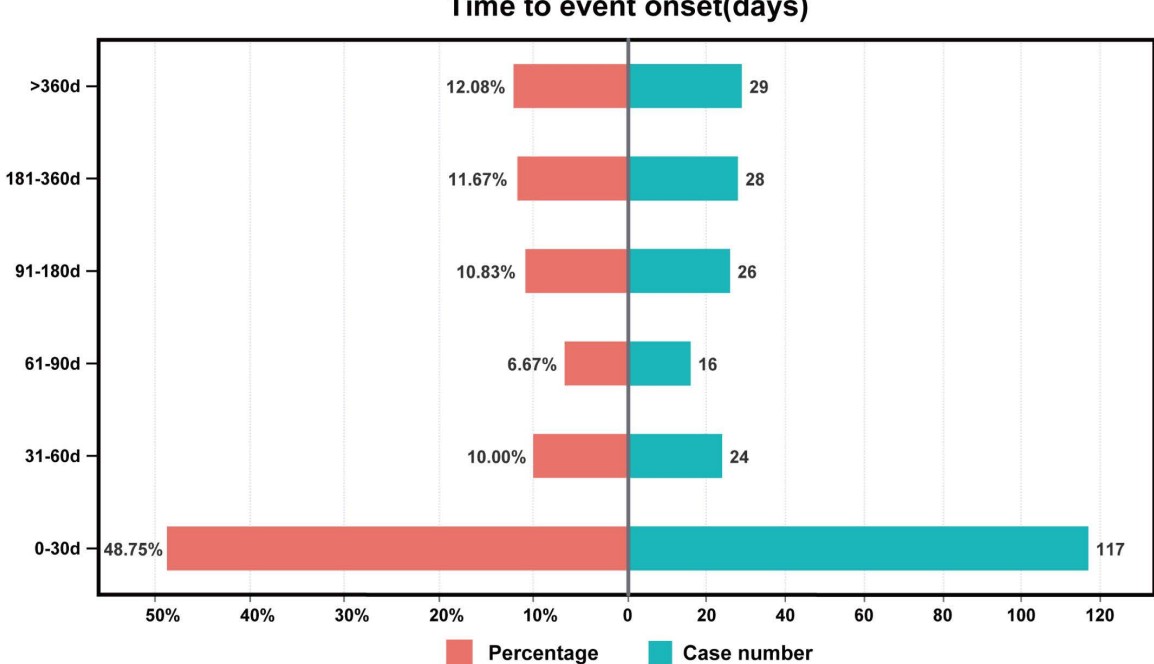

**Fig 5. Time to onset of SZC-related AEs.** SZC, sodium zirconium cyclosilicate; AEs, adverse events.

**Table 3. TTO analysis for SZC-related signals using the Weibull distribution test.**

| Cases n | TTO (days) | | Weibull distribution | | | | Failure type |
|---|---|---|---|---|---|---|---|
| | Media (IQR) | Min-Max | Scale parameter | | Shape parameter | | |
| | | | α | 95% CI | β | 95% CI | |
| 240 | 33 (5-167) | 0-838 | 95.51 | 82.83-108.19 | 0.62 | 0.54-0.70 | Early failure |

TTO, Time-to-onset; SZC, sodium zirconium cyclosilicate.

## Cardiac disorder-related AEs

Potassium levels are critical for normal cardiac function. Both hyperkalemia and hypokalemia can lead to cardiac dysfunction [23]. It is unclear whether SZC directly causes or exacerbates heart failure. However, other SZC-related AEs may potentially worsen heart failure. For example, SZC-induced fluid retention and edema may increase the blood volume, imposing cardiac stress [24], which can be particularly challenging for patients with pre-existing cardiac dysfunction. Moreover, potassium homeostasis is central to regulating cellular membrane excitability; therefore, ventricular fibrillation observed as an adverse reaction to SZC may be associated with drug-induced hypokalemia [25]. During ventricular pacing, hypokalemia activates the small conductance calcium-activated potassium channel (IKAS), reducing the action potential duration (APD) and maintaining repolarization reserve at late-activated sites. Blocking IKAS significantly prolongs APD at the late-activated sites, thereby inducing ventricular fibrillation [26]. However, hyperkalemia triggers ventricular fibrillation [27], thus indicating that SZC-induced hypokalemia is not the sole cause of arrhythmia. In summary, both SZC-induced fluid retention and the failure to maintain potassium levels within a normal range may increase the risk of cardiac disorders.

RAAS inhibitors (RAASi) are the cornerstone therapies for cardiovascular protection in patients with heart failure [28]. However, they may predispose the patients to hyperkalemia, which increases the risk of arrhythmias and sudden death [29]. Notably, SZC can enhance patient tolerance to RAASi in heart failure management, thereby improving long-term outcomes [30,31]. Therefore, physicians must carefully evaluate the patient's overall condition to determine the most beneficial treatment strategy.

## Gastrointestinal system disorder-related AEs

The analysis of PTs within the gastrointestinal system highlighted constipation, hard stools, fecaloma, gastrointestinal motility disorder, ileus, and intestinal perforation as the common AEs. Both constipation and hard stools are listed on the SZC drug label; they are common AEs of potassium binders [32,33]. SZC, a novel potassium binder, differs from traditional polymer-based binders. The volume of SZC decreases when mixed with deionized water. This unique non-polymer characteristic reduces the likelihood of constipation, compared with conventional potassium binders [34,35]. The Hyperkalemia Randomized Intervention Multi-dose ZS-9 phase III trial reported a < 10% incidence of constipation during SZC therapy [36]. A retrospective study demonstrated that older adults with hyperkalemia and CKD required fewer anti-constipation medications upon SZC therapy than individuals on conventional potassium binders. Therefore, SZC is less likely to worsen constipation [37]. Additionally, a case report described an older adult with refractory constipation caused by calcium polystyrene sulfonate; the patient experienced substantial improvement after transitioning to SZC, while maintaining normal serum potassium levels [38]. Intestinal perforation is mentioned in the SZC drug label; however, the exact risk of SZC causing intestinal perforation remains unclear. A national study demonstrated low risks of intestinal ischemia, thrombosis, or other serious gastrointestinal events; they did not differ among potassium-binding drugs [39]. Nevertheless, for patients with pre-existing gastrointestinal motility disorders, SZC use should be approached carefully to minimize the risk of severe outcomes, despite the low probability of such events.

## Metabolism and nutrition disorder-related AEs

As a potassium-lowering agent, SZC induces hypokalemia. The risk of hypokalemia has been strongly associated with the dosage and duration of SZC. A 4-week-long study reported no hypokalemia in patients treated with 5 g of SZC, whereas the probability of hypokalemia increased to approximately 10% in patients receiving 10 g or 15 g of SZC [36]. Similarly, another 1-year-long study demonstrated that despite monthly SZC dose adjustments, approximately 5% of the patients experienced reductions in serum potassium, with some developing hypokalemia during treatment [6]. Therefore, physicians should monitor serum potassium levels during SZC administration and adjust the dosage to prevent hypokalemia and related AEs.

Metabolic acidosis is not listed as an adverse reaction on the SZC drug label. This may be attributed to the underlying conditions causing hyperkalemia, particularly CKD. Metabolic acidosis or low serum bicarbonate is one of the first complications to emerge in renal failure [40]. However, it is challenging to determine whether these reactions are directly caused by SZC, co-medications, underlying health conditions, or other treatment strategies. Interestingly, previous studies on SZC have reported contrasting findings. Phase 2 clinical trials [41] and open-label trials [42,43] suggested that SZC not only fails to induce metabolic acidosis but also leads to increased serum bicarbonate and decreased serum urea levels. The SZC-associated increase in serum bicarbonate is attributed to ammonium sequestration in

the gastrointestinal tract, supporting a mechanism that involves decreased serum urea levels rather than reduced serum potassium levels or increased urine pH [44]. This mechanism has been corroborated in animal studies, such as a mouse model of CKD, where SZC sequestered ammonium in the gastrointestinal tract [45]. Increased serum bicarbonate levels after SZC therapy are relevant, particularly for patients with CKD. SZC may raise serum bicarbonate levels, thus serving as an adjunctive treatment for patients with CKD along with acidosis [46].

## General disorder-related AEs

The evidence for death as an adverse reaction to SZC is controversial. SZC may exert minimal or no effect on all-cause mortality in patients with CKD [32]. However, several factors may contribute to the high mortality real world. First, hyperkalemia is associated with increased mortality. In an observational study, the in-hospital mortality was 30% for patients with hyperkalemia [47]. Furthermore, chronic renal insufficiency and heart failure—the common causes of hyperkalemia—are independently associated with high mortality, particularly in older adults [48]. SZC effectively manages hyperkalemia symptoms; however, it neither addresses the underlying causes nor prevents the progression of primary diseases, such as CKD or heart failure. Thus, while SZC alleviates hyperkalemia, it does not modify the trajectory of comorbid conditions, which contribute to mortality.

Edema is a common adverse reaction to SZC. Most clinical trials on SZC have highlighted edema in their safety assessments. Patients treated with SZC demonstrated a 4.30 times higher risk of developing edema than the placebo group [49]. The incidence of edema is strongly associated with the mechanism of action of SZC. SZC is a highly selective cation exchanger; it captures potassium in the intestine by exchanging it with sodium and hydrogen ions [50], thus increasing sodium absorption. Moreover, sodium constitutes approximately 8% of the weight of SZC [51]. This increase in sodium absorption can result in elevated serum sodium levels, potentially leading to fluid retention and edema. Hence, physicians should monitor the risk of fluid retention during SZC therapy, particularly in patients with heart failure or CKD, to prevent exacerbation of the underlying conditions [24].

## Sex-based differences

Sex-based differences were observed for AEs. AEs were more frequent in men than in women, excluding cases with unknown gender. This finding may be attributable to the higher prevalence of hyperkalemia in men [52]. Men have lower aldosterone levels than women, which results in reduced potassium excretion [53]. Men are more likely to experience abnormal CT scans, abnormal blood potassium, scrotal edema, increased blood potassium, product preparation issues, and edema. In contrast, women were more prone to experience increased blood sodium, abnormal CT scans, abnormal blood potassium, increased blood potassium, and fecaloma. Notably, men are at a higher risk of developing edema, whereas women are at a higher risk of developing gastrointestinal disorders. This can be attributed to several theories, including slower gut transit in women because of fluctuating progesterone and estrogen levels [54–56] or pelvic floor damage related to obstetric history [57–59]. Apart from biological factors, sex-based social factors are central to these differences. Men tend to downplay illness, often delaying medical intervention, whereas women are more proactive in seeking healthcare and engaging in health-promoting behavior. This difference may contribute to the higher likelihood of severe AEs in men [60]. Our results highlight these sex-specific AEs. Despite warranting further validation, this study provides insights into improving medication monitoring for both men and women. Moreover, acknowledging sex-based social factors is crucial to enhancing safe and effective SZC therapy.

### TTO analysis

TTO refers to the duration between initiating drug administration and AE onset [61]. TTO data facilitates assessing the risk of AEs within specific timeframes, providing insights into the temporal association between drug exposure and adverse outcomes. SZC-related AEs predominantly occurred within the first 1 to 2 months of treatment, with a median onset time of 33 days. This finding underscores the importance of close monitoring, particularly after 1 month of therapy. Weibull distribution analysis is used to assess the likelihood and timing of drug-related AEs [22]. In this study, an "early failure" pattern was identified, indicating an elevated incidence of AEs immediately after initiating treatment. However, AEs persisted even up to 1 year after initiating SZC therapy. Therefore, extended follow-up periods are crucial to comprehensively identify SZC-related AEs.

This real-world observational study has several limitations. First, the FAERS is a spontaneous reporting system that relies on voluntary submissions. This feature may have introduced under-reporting, selective reporting, and incomplete data, thus compromising the robustness and reliability of AE data and leading to potential biases. Second, the disproportionality analysis primarily detects safety signal strength and establishes statistical associations. However, it does not determine risk or establish causality between drug exposure and AEs. Therefore, well-designed clinical studies are needed to investigate causal associations. Third, unaccounted variables—such as the underlying disease severity, coexisting medical conditions, and polypharmacy—complicate the efforts to control for confounding factors [62]. Finally, the total number of patients receiving SZC therapy was unknown. Therefore, the incidence of each SZC-related AE could not be quantified. Nonetheless, utilizing the FAERS in pharmacovigilance facilitates accessing a real-world dataset, enabling the early detection of drug safety signals. This study contributes valuable insights into the safe use of medications and informs further clinical research.

## Conclusion

In conclusion, this pharmacovigilance analysis of the FAERS scientifically and systematically quantified the potential risks associated with SZC therapy. Physicians should carefully monitor all populations and determine the risk of AEs. However, this study necessitates considering the inherent limitations of the FAERS, along with potential confounders and biases. Thus, the analysis results should be interpreted cautiously. The insights offer valuable evidence to guide further investigations and inform clinical practice regarding SZC therapy.

## Supporting information

**S1 Table. Contingency table for AE signal detection.**
(DOCX)

**S2 Table. Four methods, formulas and thresholds.**
(DOCX)

**S3 Table. Signal strength of reports of SZC at the SOC level in the FAERS database.**
(DOCX)

**S4 Table. The signal strength of AEs at PT level in male subgroup using ROR, PRR, BCPNN, and EBGM.**
(DOCX)

**S5 Table. The signal strength of AEs at PT level in female subgroup using ROR, PRR, BCPNN, and EBGM.**
(DOCX)

**S6 File. The READUS checklist for abstract and manuscript.**
(DOCX)

## Acknowledgments

We would like to thank the Food and Drug Administration Adverse Event Reporting System and all those who participated in this study. We thank the CNSknowall platform (https://cnsk-nowall.com) for providing data analysis services.

## Author contributions

**Conceptualization:** Yongfei Yu, Kaiyu Zhang, Enchao Zhou.

**Data curation:** Yongfei Yu, Kaiyu Zhang.

**Formal analysis:** Kaiyu Zhang.

**Funding acquisition:** Enchao Zhou.

**Investigation:** Yongfei Yu.

**Methodology:** Chen Yong.

**Project administration:** Enchao Zhou.

**Resources:** Kaiyu Zhang.

**Supervision:** Enchao Zhou.

**Validation:** Guoshun Huang, Yuan Wei.

**Visualization:** Yongfei Yu, Jinglin Gao, Yuan Wei.

**Writing – original draft:** Yongfei Yu, Kaiyu Zhang, Jinglin Gao.

**Writing – review & editing:** Guoshun Huang, Chen Yong, Enchao Zhou.

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
