## [Decision Letter · Decision Letter 0]

7 Feb 2025

PONE-D-24-48785Safety assessment of sodium zirconium cyclosilicate: a FAERS-based disproportionality analysisPLOS ONE

Dear Dr. Zhou,

Thank you for submitting your manuscript to PLOS ONE. After careful consideration, we feel that it has merit but does not fully meet PLOS ONE’s publication criteria as it currently stands. Therefore, we invite you to submit a revised version of the manuscript that addresses the points raised during the review process.

We look forward to receiving your revised manuscript.

Kind regards,

Nik Hisamuddin Nik Ab. Rahman

Academic Editor

PLOS ONE

Journal Requirements:

“Our research was supported by the National Natural Science Foundation of China under grant 82474427 to EZ, Jiangsu Natural Science Foundation of China under grant BE2023791 to EZ, Jiangsu Province Leading Talents Cultivation Project for Traditional Chinese Medicine under grant SLJ0319 to EZ, Jiangsu Province Traditional Chinese Medicine Science and Technology Project under grant ZD202007 to EZ and Postgraduate Research & Practice Innovation Program of Jiangsu Province under grant SJCX_0724 to YY.”

Reviewers' comments:

Reviewer's Responses to Questions

**Comments to the Author**

1. Is the manuscript technically sound, and do the data support the conclusions?

Reviewer #1: Yes

Reviewer #2: Yes

2. Has the statistical analysis been performed appropriately and rigorously? 

Reviewer #1: Yes

Reviewer #2: I Don't Know

3. Have the authors made all data underlying the findings in their manuscript fully available?

Reviewer #1: Yes

Reviewer #2: Yes

4. Is the manuscript presented in an intelligible fashion and written in standard English?

Reviewer #1: Yes

Reviewer #2: Yes

5. Review Comments to the Author

Reviewer #1: The study investigates adverse events of SZC using FARES data base. The methodology and analysis are appropriate and the authors have recognised the limitations and potential biases. I have a few comments related to the methodology.

1) The authors may wish to expand on how they dealt with duplicate entries and verified each report related SCZ.

2) It is stated that 8,067,923 AE reports were collected from the FAERS database during the study period and after rigorous data cleaning, 1,154 SZC-related AEs were identified for analysis. The authors may wish to provide details of the so called data cleaning” process, especially with respect to reports related SZC. ( number of reports related to SCZ excluded, the reasons for exclusion etc.)

Reviewer #2: In the manuscript “Safety Assessment of Sodium Zirconium Cyclosilicate: A FAERS-Based Disproportionality Analysis” Yu et al. describe a retrospective study analyzing adverse effects reported in patients who used sodium zirconium cyclosilicate (SZC), which since 2018 has been a therapeutic agent for the treatment of hyperkalemia. The study analyzes a large number of real-world reports recorded in the Food and Drug Administration Adverse Event Reporting System (FAERS) database between 2018 and 2023. Through statistical tests with disproportionality measures, the authors obtained relevant information quantifying the number of adverse event reports. This pharmacovigilance strategy is interesting because traditional qualitative analyzes of large databases are expensive and require long execution times. Thus, quantitative analysis (data mining) represents a faster and more efficient pharmacovigilance tool.

Through this strategy, the authors confirmed the known adverse effects of SZC and identified new observations of side effects, covering 21 organ systems. They also highlighted the need for careful clinical monitoring and further studies to confirm associations of reported adverse effects with SZC, as disproportionality analysis does not prove causality.

The authors have produced an interesting manuscript containing original and relevant data. There are minor issues that need to be addressed before it is accepted for publication on Plos One.

Minor Issues

References: The way references are cited needs to be revised to be presented in the same standard. Citations from periodicals are sometimes written in full and in capital letters, sometimes in abbreviated form and in lowercase letters.

6. PLOS authors have the option to publish the peer review history of their article (what does this mean? ). If published, this will include your full peer review and any attached files.

**Do you want your identity to be public for this peer review?** For information about this choice, including consent withdrawal, please see our Privacy Policy .

Reviewer #1: No

Reviewer #2: **Yes: ** MAURICIO YOUNES-IBRAHIM

---

## [Author Response · Author response to Decision Letter 0]

11 Feb 2025

Dear Editor and Reviewers,

Thank you for perspective on our work. Your comments have been extremely helpful for improving the quality of our manuscript. We have reviewed and revised the reference list. Now we ensure that it is complete and correct. Words shown in red are the changes we have made in the manuscript. We have also provided our point-by-point response to the comments from each reviewer, and we have highlighted the changes in the revised manuscript. The complete details of each file is listed. We sincerely hope that you find our responses and modifications satisfactory and consider the manuscript suitable for publication.

Response to academic editor:

[1)Thank you for stating the following financial disclosure:

“Our research was supported by the National Natural Science Foundation of China under grant 82474427 to EZ, Jiangsu Natural Science Foundation of China under grant BE2023791 to EZ, Jiangsu Province Leading Talents Cultivation Project for Traditional Chinese Medicine under grant SLJ0319 to EZ, Jiangsu Province Traditional Chinese Medicine Science and Technology Project under grant ZD202007 to EZ and Postgraduate Research & Practice Innovation Program of Jiangsu Province under grant SJCX_0724 to YY.”

Please include this amended Role of Funder statement in your cover letter; we will change the online submission form on your behalf.]

Response: Thank you for your reminder. We have confirmed that the funders had no role in study design, data collection and analysis, decision to publish, or preparation of the manuscript. The statement has been added in the manuscript and cover letter.

[2)While revising your submission, please upload your figure files to the Preflight Analysis and Conversion Engine (PACE) digital diagnostic tool, https://pacev2.apexcovantage.com/. PACE helps ensure that figures meet PLOS requirements. To use PACE, you must first register as a user. Registration is free. Then, login and navigate to the UPLOAD tab, where you will find detailed instructions on how to use the tool. If you encounter any issues or have any questions when using PACE, please email PLOS at figures@plos.org. Please note that Supporting Information files do not need this step.]

Response: Thank you for your helpful guidance. The figures have been uploaded to the Preflight Analysis and Conversion Engine (PACE) digital diagnostic tool. PACE generated figure files have been uploaded with the revised submission.

Response to the reviewer#1’s comments:

[1)The authors may wish to expand on how they dealt with duplicate entries and verified each report related SCZ.]

Response: Thank you for your detailed review of our method. The drug name was standardized through the Medex_UIMA_1.8.3 system. Reports suspecting Sodium zirconium cyclosilicate as the primary drug associated with AEs were extracted. The FDA-recommended criteria were followed to remove duplicate entries[1, 2]. For identical CASEIDs, the latest FDA_DT was selected; for similar CASEID and FDA_DT, the record with the higher PRIMARYID was selected[3].

We have refined this part in the data source and data processing section of the manuscript.

References:

1. Shu Y, He X, Liu Y, Wu P, Zhang Q. A Real-World Disproportionality Analysis of Olaparib: Data Mining of the Public Version of FDA Adverse Event Reporting System. Clin Epidemiol. 2022;14:789-802. https://10.2147/CLEP.S365513 PMID: 35789689

2. Wang Y, Zhao B, Yang H, Wan Z. A real-world pharmacovigilance study of FDA adverse event reporting system events for sildenafil. Andrology. 2024;12(4):785-92. https://10.1111/andr.13533 PMID: 37724699

3. Cui Z, Cheng F, Wang L, Zou F, Pan R, Tian Y, et al. A pharmacovigilance study of etoposide in the FDA adverse event reporting system (FAERS) database, what does the real world say? Front Pharmacol. 2023;14:1259908. https://10.3389/fphar.2023.1259908 PMID: 37954852

[2) It is stated that 8,067,923 AE reports were collected from the FAERS database during the study period and after rigorous data cleaning, 1,154 SZC-related AEs were identified for analysis. The authors may wish to provide details of the so called data cleaning” process, especially with respect to reports related SZC. ( number of reports related to SCZ excluded, the reasons for exclusion etc.)]

Response: Thank you for reading the manuscript so carefully. Considering the drug approval timeline, the FAERS database's DEMO module initially contained 9,440,084 reports from the third quarter of 2018 to the fourth quarter of 2023. Following FDA-recommended deduplication protocols, we removed 1,372,161 duplicate entries, retaining 8,067,923 standardized DEMO records. Subsequently, we conducted systematic searches in the DRUG dataset using standardized medication nomenclature, establishing cross-database linkages through the primaryid field. This identifier enabled targeted extraction of 1,154 case records containing demographic and administrative information related to adverse events from the DEMO database. These validated primaryid entries were further utilized to retrieve 2,027 specific AE reports from the REAC database. The consolidated dataset was then subjected to integrative analysis and critical evaluation.

The absence of granular clinical metadata precluded comprehensive identification and exclusion of data susceptible to potential bias, constituting a fundamental methodological constraint in FAERS-related investigation. Therefore, well-designed clinical studies are needed in the future to investigate causal associations.

Response to the reviewer#2’s comments:

[Minor Issues References: The way references are cited needs to be revised to be presented in the same standard. Citations from periodicals are sometimes written in full and in capital letters, sometimes in abbreviated form and in lowercase letters.]

Response: Thank you very much for your detailed evaluation of our research. We have reviewed and revised the reference list. We have replaced the journal names in the references by substituting the original full journal titles with standardized abbreviated forms that meet the required guidelines, and have additionally supplemented each reference entry with its corresponding PMID number. We also replaced "doi:" with "https//" to enhance compliance with the journal's formatting guidelines. Now we ensure that it is complete and correct. The changes in references are as follows:

1. Kim MJ, Valerio C, Knobloch GK. Potassium Disorders: Hypokalemia and Hyperkalemia. AMERICAN FAMILY PHYSICIAN. 2023;107(1):59-70.

We have replaced "AMERICAN FAMILY PHYSICIAN" with "Am Fam Physician" and have added "PMID: 36689973" to complete the information of this reference.

2.Nilsson E, Gasparini A, Arnlov J, Xu H, Henriksson KM, Coresh J, et al. Incidence and determinants of hyperkalemia and hypokalemia in a large healthcare system. International Journal of Cardiology. 2017;245:277-84. doi: 10.1016/j.ijcard.2017.07.035.

We have replaced "International Journal of Cardiology" with "Int J Cardiol" and have added "PMID: 28735756" to complete the information of this reference.

3.Weiss JN, Qu Z, Shivkumar K. Electrophysiology of Hypokalemia and Hyperkalemia. Circulation-Arrhythmia and Electrophysiology. 2017;10(3). doi: 10.1161/circep.116.004667.

We have replaced "Circulation-Arrhythmia and Electrophysiology" with "Circ Arrhythm Electrophysiol" and have added "PMID: 28314851" and "e004667" to complete the information of this reference.

4.Llubani R, Vukadinovic D, Werner C, Marx N, Zewinger S, Bohm M. Hyperkalaemia in Heart Failure-Pathophysiology, Implications and Therapeutic Perspectives. Current heart failure reports. 2018;15(6):390-7. doi: 10.1007/s11897-018-0413-9.

We have replaced "Current heart failure reports" with "Curr Heart Fail Rep" and have added "PMID: 30421355" to complete the information of this reference.

5.Hoy SM. Sodium Zirconium Cyclosilicate: A Review in Hyperkalaemia. DRUGS. 2018;78(15):1605-13. doi: 10.1007/s40265-018-0991-6.

We have replaced "DRUGS" with "Drugs" and have added "PMID: 30306338" to complete the information of this reference.

6.Spinowitz BS, Fishbane S, Pergola PE, Roger SD, Lerma EV, Butler J, et al. Sodium Zirconium Cyclosilicate among Individuals with Hyperkalemia A 12-Month Phase 3 Study. CLINICAL JOURNAL OF THE AMERICAN SOCIETY OF NEPHROLOGY. 2019;14(6):798-809. doi: 10.2215/CJN.12651018.

We have replaced "CLINICAL JOURNAL OF THE AMERICAN SOCIETY OF NEPHROLOGY" with "Clin J Am Soc Nephrol" and have added "PMID: 31110051" to complete the information of this reference.

7.Jiang Y, Zhou L, Shen Y, Zhou Q, Ji Y, Zhu H. Safety assessment of Brexpiprazole: Real-world adverse event analysis from the FAERS database. JOURNAL OF AFFECTIVE DISORDERS. 2024;346:223-9. doi: 10.1016/j.jad.2023.11.025.

We have replaced "JOURNAL OF AFFECTIVE DISORDERS" with "J Affect Disord" and have added "PMID: 37956832" to complete the information of this reference.

8.Shu Y, Ding Y, Liu Y, Wu P, He X, Zhang Q. Post-Marketing Safety Concerns With Secukinumab: A Disproportionality Analysis of the FDA Adverse Event Reporting System. FRONTIERS IN PHARMACOLOGY. 2022;13. doi: 10.3389/fphar.2022.862508.

We have replaced "FRONTIERS IN PHARMACOLOGY" with "Front Pharmacol" and have added "PMID: 35754494" and "2022;13:862508"to complete the information of this reference.

9.Wang F, Zhou B, Sun H, Wu X. Proarrhythmia associated with antiarrhythmic drugs: a comprehensive disproportionality analysis of the FDA adverse event reporting system. FRONTIERS IN PHARMACOLOGY. 2023;14. doi: 10.3389/fphar.2023.1170039.

We have replaced "FRONTIERS IN PHARMACOLOGY" with "Front Pharmacol" and have added "PMID: 37251345" and "2023;14:1170039" to complete the information of this reference.

10.Sakaeda T, Tamon A, Kadoyama K, Okuno Y. Data Mining of the Public Version of the FDA Adverse Event Reporting System. International Journal of Medical Sciences. 2013;10(7):796-803. doi: 10.7150/ijms.6048.

We have replaced "International Journal of Medical Sciences" with "Int J Med Sci" and have added "PMID: 23794943" to complete the information of this reference.

11.Shu Y, He X, Liu Y, Wu P, Zhang Q. A Real-World Disproportionality Analysis of Olaparib: Data Mining of the Public Version of FDA Adverse Event Reporting System. CLINICAL EPIDEMIOLOGY. 2022;14:789-802. doi: 10.2147/CLEP.S365513.

We have replaced "CLINICAL EPIDEMIOLOGY" with "Clin Epidemiol" and have added "PMID: 35789689" to complete the information of this reference.

12.Wang Y, Zhao B, Yang H, Wan Z. A real-world pharmacovigilance study of FDA adverse event reporting system events for sildenafil. ANDROLOGY. 2024;12(4):785-92. doi: 10.1111/andr.13533.

We have replaced "ANDROLOGY" with "Andrology" and have added "PMID: 37724699" to complete the information of this reference.

13.Cui Z, Cheng F, Wang L, Zou F, Pan R, Tian Y, et al. A pharmacovigilance study of etoposide in the FDA adverse event reporting system (FAERS) database, what does the real world say? FRONTIERS IN PHARMACOLOGY. 2023;14. doi: 10.3389/fphar.2023.1259908.

We have replaced "FRONTIERS IN PHARMACOLOGY" with "Front Pharmacol" and have added "PMID:37954852" and "2023;14:1259908" to complete the information of this reference.

14.Singh J. International conference on harmonization of technical requirements for registration of pharmaceuticals for human use. Journal of pharmacology & pharmacotherapeutics. 2015;6(3):185-7. doi: 10.4103/0976-500x.162004.

We have replaced "Journal of pharmacology & pharmacotherapeutics" with "J Pharmacol Pharmacother" and have added "PMID: 26312010" to complete the information of this reference.

15.Brown EG. Methods and pitfalls in searching drug safety databases utilising the Medical Dictionary for Regulatory Activities (MedDRA). Drug Safety. 2003;26(3):145-58. doi: 10.2165/00002018-200326030-00002.

We have replaced "Drug Safety" with "Drug Saf" and have added "PMID: 12580645" to complete the information of this reference.

16.Zhou Q, Du Z, Qu K, Shen Y, Jiang Y, Zhu H, et al. Adverse events of epidiolex: A real-world drug safety surveillance study based on the FDA adverse event reporting system (FAERS) database. ASIAN JOURNAL OF PSYCHIATRY. 2023;90. doi: 10.1016/j.ajp.2023.103828.

We have replaced "ASIAN JOURNAL OF PSYCHIATRY" with "Asian J Psychiatr" and have added "PMID: 37949044" and "2023;90:103828"to complete the information of this reference.

17.Steinhart AH, Hemphill D, Greenberg GR. Sulfasalazine and mesalazine for the maintenance therapy of Crohn's disease: a meta-analysis. The American journal of gastroenterology. 1994;89(12):2116-24.

We have replaced "The American journal of gastroenterology" with "Am J Gastroenterol" and have added "PMID: 7977225" to complete the information of this reference.

18.van Puijenbroek EP, Bate A, Leufkens HGM, Lindquist M, Orre R, Egberts ACG. A comparison of measures of disproportionality for signal detection in spontaneous reporting systems for adverse drug reactions. Pharmacoepidemiology and Drug Safety. 2002;11(1):3-10. doi: 10.1002/pds.668.

We have replaced "Pharmacoepidemiology and Drug Safety" with "Pharmacoepidemiol Drug Saf" and have added "PMID: 11998548" to complete the information of this reference.

19.Song Y, Xu Y-l, Lin Y, Zhao B, Sun Q. Fractures due to Aromatase Inhibitor Therapy for Breast Cancer: A Real-World Analysis of FAERS Data in the Past 15 Years. Oncology Research and Treatment. 2020;43(3):96-102. doi: 10.1159/000505376.

We have replaced "Oncology Research and Treatment" with "Oncol Res Treat" and have added "PMID: 31945768" to complete the information of this reference.

20.Cornelius VR, Sauzet O, Evans SJW. A Signal Detection Method to Detect Adverse Drug Reactions Using a Parametric Time-to-Event Model in Simulated Cohort Data. Drug Safety. 2012;35(7):599-610.

We have replaced "Drug Safety" with "Drug Saf" and have added "PMID: 22702641" to complete the information of this reference.

21.Kinoshita S, Hosomi K, Yokoyama S, Takada M. Time-to-onset analysis of amiodarone-associated thyroid dysfunction. Journal of Clinical Pharmacy and Therapeutics. 2020;45(1):65-71. doi: 10.1111/jcpt.13024.

We have replaced "Journal of Clinical Pharmacy and Therapeutics" with "J Clin Pharm Ther" and have added "PMID: 31400296" to complete the information of this reference.

22.Mazhar F, Battini V, Gringeri M, Pozzi M, Mosini G, Marran AMN, et al. The impact of anti-TNFα agents on weight-related changes: new insights from a real-world pharmacovigilance study using the FDA adverse event reporting system (FAERS) database. Expert Opinion on Biological Therapy. 2021;21(9):1281-90. doi: 10.1080/14712598.2021.1948529.

We have replaced "Expert Opinion on Biological Therapy" with "Expert Opin Biol Ther" and have added "PMID: 34191656" to complete the information of this reference.

23.Aldahl M, Jensen A-SC, Davidsen L, Eriksen MA, Hansen SM, Nielsen BJ, et al. Associations of serum potassium levels with mortality in chronic heart failure patients. European Heart Journal. 2017;38(38):2890-6. doi: 10.1093/eurheartj/ehx460.

We have replaced "European Heart Journal" with "Eur Heart J" and have added "PMID: 29019614" to complete the information of this reference.

24.Desai NR, Kammerer J, Budden J, Olopoenia A, Tysseling A, Gordon A. The Association of Heart Failure and Edema Events between Patients Initiating SZC or Patiromer. Kidney360. 2024;10.34067/kid.0000000586. doi: 10.34067/kid.0000000586.

We have replaced "2024;10.34067/kid.0000000586" with "2024;5(12):1835-43" to complete the information of this reference.

25.Thu Kyaw M, Maung ZM. Hypokalemia-Induced Arrhythmia: A Case Series and Literature Review. Cureus. 2022;14(3):e22940-e. doi: 10.7759/cureus.22940.

We have added "PMID: 35411269" to complete the information of this reference.

26.Chan YH, Tsai WC, Ko JS, Yin DC, Chang PC, Rubart M, et al. Small-Conductance Calcium-Activated Potassium Current Is Activated During Hypokalemia and Masks Short-Term Cardiac Memory Induced by Ventricular Pacing. Circulation. 2

---

## [Editor Report · Decision Letter 1]

21 Feb 2025

Safety assessment of sodium zirconium cyclosilicate: a FAERS-based disproportionality analysis

PONE-D-24-48785R1

Dear En-chao Zhou ,

We’re pleased to inform you that your manuscript has been judged scientifically suitable for publication and will be formally accepted for publication once it meets all outstanding technical requirements.

Kind regards,

Nik Hisamuddin Nik Ab. Rahman

Academic Editor

PLOS ONE
---

## [Editor Report · Acceptance letter]

PONE-D-24-48785R1

PLOS ONE

Dear Dr. Zhou,

I'm pleased to inform you that your manuscript has been deemed suitable for publication in PLOS ONE. Congratulations! Your manuscript is now being handed over to our production team.

Kind regards,

on behalf of

Professor Dr Nik Hisamuddin Nik Ab. Rahman

Academic Editor

PLOS ONE